# First evidence for the evolution of host manipulation by tumors during the long-term vertical transmission of tumor cells in *Hydra oligactis*

**Justine Boutry[1]\*, Océane Rieu[1], Lena Guimard[1], Jordan Meliani[1], Aurora M Nedelcu[2], Sophie Tissot[1], Nikita Stepanskyy[1], Beata Ujvari[1,3], Rodrigo Hamede[4], Antoine M Dujon[1,3], Jácint Tökölyi[5], Fréderic Thomas[1]**

[1]CREEC/CANECEV (CREES), MIVEGEC, Unité Mixte de Recherches, IRD 224-CNRS 5290 Université de Montpellier, Montpellier, France; [2]Department of Biology, University of New Brunswick, Fredericton, Canada; [3]School of Life and Environmental Sciences, Deakin University, Waurn Ponds, Australia; [4]School of Biological Sciences, University of Tasmania, Hobart, Australia; [5]MTA-DE "Momentum" Ecology, Evolution and Developmental Biology Research Group, Department of Evolutionary Zoology, University of Debrecen, Debrecen, Hungary

**\*For correspondence:**
justine.boutry@ird.fr

## eLife Assessment

This interesting study explores whether tumor cells can manipulate their *Hydra* hosts and includes **important** findings on the consequences for the fitness of the host *Hydra*. The evidence supporting these findings is **convincing**. The work will be of broad interest to many fields including development biology, evolutionary biology and tumor biology.

**Abstract** While host phenotypic manipulation by parasites is a widespread phenomenon, whether tumors, which can be likened to parasite entities, can also manipulate their hosts is not known. Theory predicts that this should nevertheless be the case, especially when tumors (neoplasms) are transmissible. We explored this hypothesis in a cnidarian *Hydra* model system, in which spontaneous tumors can occur in the lab, and lineages in which such neoplastic cells are vertically transmitted (through host budding) have been maintained for over 15 years. Remarkably, the hydras with long-term transmissible tumors show an unexpected increase in the number of their tentacles, allowing for the possibility that these neoplastic cells can manipulate the host. By experimentally transplanting healthy as well as neoplastic tissues derived from both recent and long-term transmissible tumors, we found that only the long-term transmissible tumors were able to trigger the growth of additional tentacles. Also, supernumerary tentacles, by permitting higher foraging efficiency for the host, were associated with an increased budding rate, thereby favoring the vertical transmission of tumors. To our knowledge, this is the first evidence that, like true parasites, transmissible tumors can evolve strategies to manipulate the phenotype of their host.

## Introduction

It is now widely established that animals (metazoans) are not autonomous entities, but rather holobionts composed of the host cell community, various commensal and mutualistic microorganisms, a

**eLife digest** Parasites live on or inside host organisms and rely on them for survival. To increase their chances of surviving and spreading, many parasites manipulate their hosts, changing characteristics such as how the host looks, behaves or reproduces. Abnormal tissues growths (known as tumors) can also manipulate their host to fulfil their own needs. For example, some tumors trigger growth of new blood vessels to increase their access to oxygen, nutrients, and to enable them to spread to other organs in a process known as metastasis. In rare cases, tumors can even spread between hosts, much like parasites.

The tiny freshwater animal *Hydra oligactis* can develop long-term tumors that are passed from one generation to the next. Intriguingly, hydras with these tumors grow more tentacles than those without, helping them to catch more food. This increases their ability to reproduce, which could potentially help to pass tumors on to other hydra hosts.

Boutry et al. aimed to explore whether long-term tumors are directly responsible for increasing the number of hydra tentacles and how they affect tumor transmission. To do so, the researchers tracked tentacle development in hydra with long-term tumors that are passed down over generations and those with spontaneous tumors, which are not transmissible. This revealed that hydras with long-term transmissible tumors develop significantly more tentacles than their healthy counterparts. Notably, hydras with spontaneously occurring tumors did not exhibit this trait.

Next, Boutry et al. transplanted tumor tissues from hydra with long-term tumors or spontaneous tumors onto healthy hydra. Only the long-term transmissible tumor tissue triggered additional tentacle development in previously healthy hosts. This suggests that the tumors have evolved the ability to manipulate host tissue development. Moreover, extra tentacles significantly improved feeding efficiency, leading to higher reproductive rates. Since hydra reproduce by producing genetically identical clones, a higher reproductive output translates into more opportunities for tumor cells to spread.

The findings of Boutry et al. provide the first evidence that transmissible tumors can actively reshape their host's body structure to benefit their own transmission, suggesting they may have evolved sophisticated strategies to persist and proliferate. Future research could investigate the genetic and biochemical mechanisms underlying this ability. Understanding these processes may offer new insights into the evolutionary dynamics of tumors and identify any similarities with parasitism.

---

diversity of parasitic taxa, and in many cases, tumor cells (*Ujvari et al., 2016a*; *Dheilly et al., 2019*). While it has been shown that parasites and mutualistic organisms have frequently evolved the ability to manipulate the host phenotype (e.g., reproduction, behavior, morphology, physiology, body odor; *Hughes et al., 2012*, *Johnson and Foster, 2018*; *Naundrup et al., 2022*) to favor their transmission and/or survival, sometimes inducing dramatic alterations (*Doherty, 2020*), this phenomenon has been poorly investigated in the context of host–tumor interactions. The only unambiguous evidence of such manipulation is found in the tumoral microenvironment when, for instance, malignant cells manipulate the surrounding healthy cells to induce the growth of blood vessels that bring oxygen and nutrients to the tumor, a process called neoangiogenesis (*Hicklin and Ellis, 2005*). Evidence regarding the ability of tumor cells to manipulate their host phenotype at other levels is scarce (see *Tissot et al., 2016* for a review), and this probably reflects a real rarity of this phenomenon. Indeed, host manipulation capabilities often rely on sophisticated and complex adaptations that evolve during long coevolutionary processes, whereas tumors (neoplasms) most often evolve for only a few years and then die with their host without being able to transmit any acquired traits (*Arnal et al., 2015*). However, like 'true parasites', transmissible tumor cells (vertically – such as in *Hydra oligactis* [*Domazet-Lošo et al., 2014*; *Boutry et al., 2022b*]; or horizontally – such as in Tasmanian devils [*Jones et al., 2008*]) could more easily evolve manipulative abilities (*Tissot et al., 2016*; *Ujvari et al., 2016b*). Nevertheless, to date, we lack evidence that transmissible cancers can manipulate the phenotype of their hosts.

*Hydra oligactis* are cnidarians that frequently develop spontaneous tumors in the lab. These tumors correspond to abnormal proliferation of germline cells in the hydra's ectoderm (*Boutry et al., 2023*). In 2014, *Domazet-Lošo et al., 2014* described a fascinating case of two hydra lineages harboring neoplasms capable of vertical transmission through buds, during the asexual reproduction of hydras. Thanks to this transmission, these two tumorous hydra lines have been maintained under laboratory

conditions for more than 15 years now (*Lange, 2010*). Beyond the presence of large tumor masses, a curious feature of these hydras is their increased number of tentacles. Specifically, while healthy polyps typically have 6–7 tentacles, tumorous polyps have between 8 and 13 tentacles, even up to 20 (*Domazet-Lošo et al., 2014*). This phenotype is particularly impressive given that tentacle numbers in hydras usually vary in a very small range (from 5 to 8 tentacles, depending on temperature and body size) (*Shostak, 2012*). Recently, *Boutry et al., 2022a* demonstrated that tumorous hydras featuring these supernumerary tentacles capture more prey than healthy ones, regardless of whether prey is rare or abundant in the environment. In contrast, wild specimens of *H. oligactis* developing spontaneous tumors when placed in laboratory conditions do not seem to develop additional tentacles (*Domazet-Lošo et al., 2014*). Since food intake, survival, as well as budding rate are usually positively correlated in hydras (*Tökölyi et al., 2016*), the presence of additional tentacles in certain tumorous hydras is likely to have an adaptive value, favoring host survival despite the cost of bearing tumors, and this can ultimately enhance the hydra's reproductive potential and/or the tumor cells' transmission (see also *Boutry et al., 2022b*).

However, whether the host or the tumor is responsible for the alteration of the tentacles number is not clear yet. Furthermore, in one *H. oligactis* lineage, called the St. Petersburg strain, tumor initiation and maintenance require the presence of a specific microbiome (*Rathje et al., 2020*), suggesting that bacteria may also be involved in the phenotypic alteration of the host to promote their own transmission. Here, through a series of experiments using a set of healthy and tumor tissue grafts (*Figure 1*) from tumors associated with different phenotypes, we attempted to determine whether the host or the tumor is responsible for the growth of additional tentacles in the tumorous hydra.

## Results

### Tentacle numbers in hydras bearing transmitted versus spontaneous tumors

As expected (see 'Introduction'), hydras from both strains harboring transmissible tumors had an enhanced number of tentacles, with typically more than four additional tentacles in both the Rob (*Figure 2A*, control, $\tilde{x}$=6[6–6] (median, first, and third quartiles); transmissible tumors $\tilde{x}$=10[9–12], p<0.001) and the SpB (*Figure 2A*, control $x$ = 6[5.5–6]; transmissible tumors $\tilde{x}$ = 10[8–10], p<0.001) strains. In contrast, SpB hydras bearing spontaneous tumors did not show an increased number of tentacles compared to healthy controls (*Figure 2A*, control $\tilde{x}$=6[5.5–6]; spontaneous tumors $x$=6[6–6.5], p=0.371). However, Mt hydras bearing spontaneous tumors showed an increase in the number of tentacles compared with controls, but of lower magnitude than for Rob and SpB strains with transmissible tumors, with typically one additional tentacle (*Figure 2A*, control $x$ = 6[5–6]; spontaneous tumors $\tilde{x}$=7[6–7.5], p<0.001).

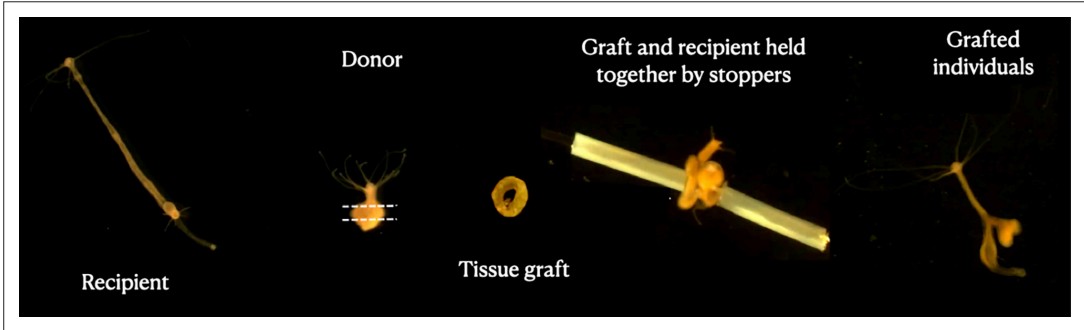

**Figure 1.** Grafting protocol. Images represent each step of the transplantation process, starting with the recipient polyp (left); the donor is then cut into a ring-shaped tissue graft that is inserted into a needle, put in contact with the recipient tissue and blocked by stopping tubes. At the end (right), a transplanted individual can be seen 3 days after the transplant.

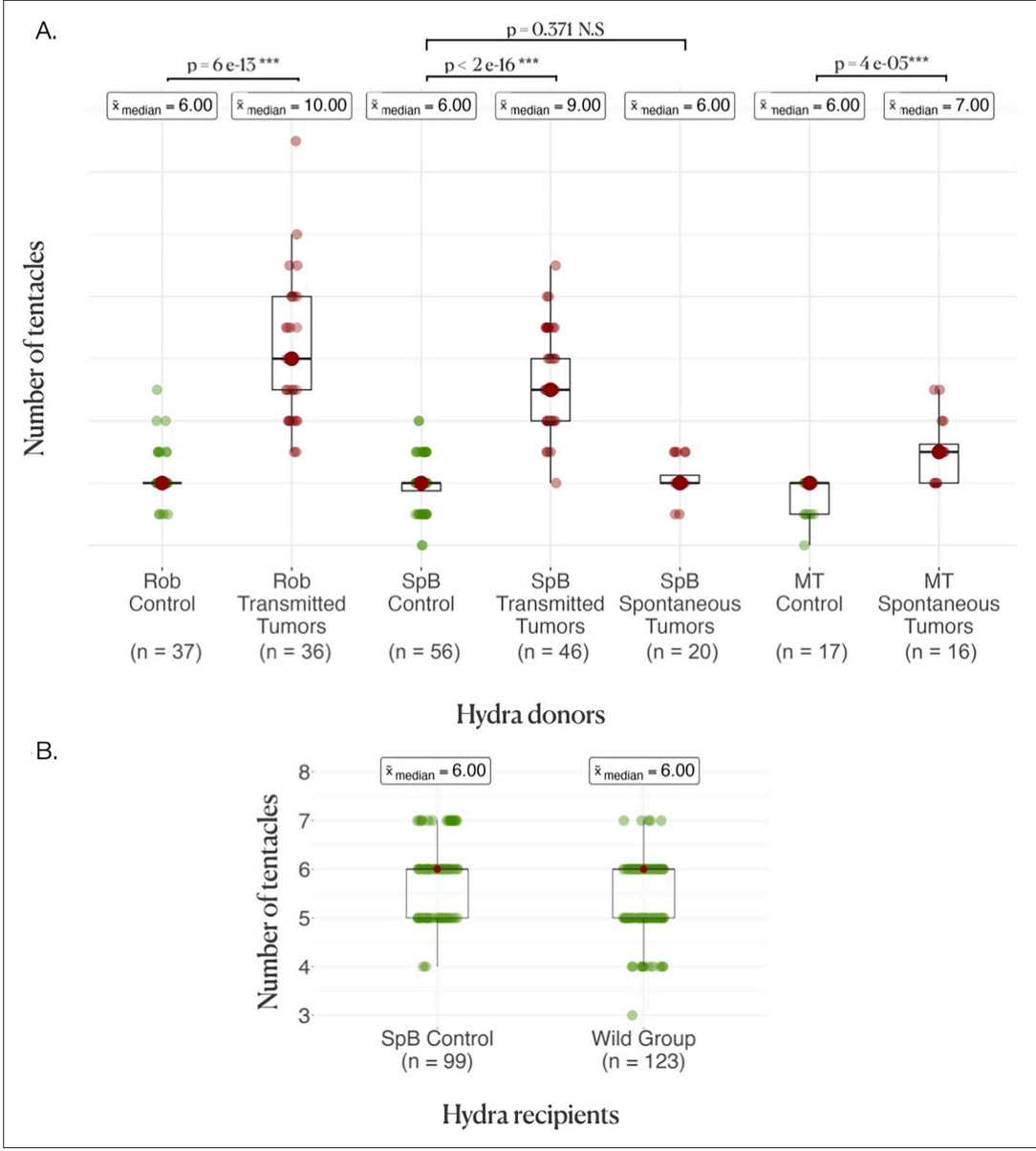

**Figure 2.** Donor and recipient tentacle phenotype. (**A**) Boxplots represent the number of tentacles in each donor line; the average number of tentacles in each group is written above in the boxes, and the rows represent the comparison between the control and the tumor medians in each line, with the p-values of the corresponding Wilcoxon–Mann–Whitney tests. (**B**) Boxplots represent the number of tentacles in different groups of recipients.

## Tumor development following tissue transplantation

On average, healthy hydras grafted with tumorous tissue, either from the transmissible or the spontaneous tumors, have an increased risk to subsequently develop tumors compared to those grafted with healthy tissue (*Figure 3*, pie charts). Hydras grafted with transmissible tumors developed on average eight times more tumors (incidence rate ratio [IRR]=8.00 [2.73–23.49], p<0.001) than those transplanted with healthy tissue, whereas for spontaneous tumor transplants the risk only increased 2.61 times (IRR = 2.61 [1.09–6.23], p=0.031). Nevertheless, for the transplantation of transmissible tumors from Rob donors on SpB healthy recipients, the effect appears weaker (see *Figure 3A*, pie charts), although it is not detected as significantly different from the other transplant combinations (see supplementary section 3). However, an intriguing result is that hydra grafted with healthy tissues also developed tumors but at lower, yet non-negligible, rate compared to hydra grafted with tumorous

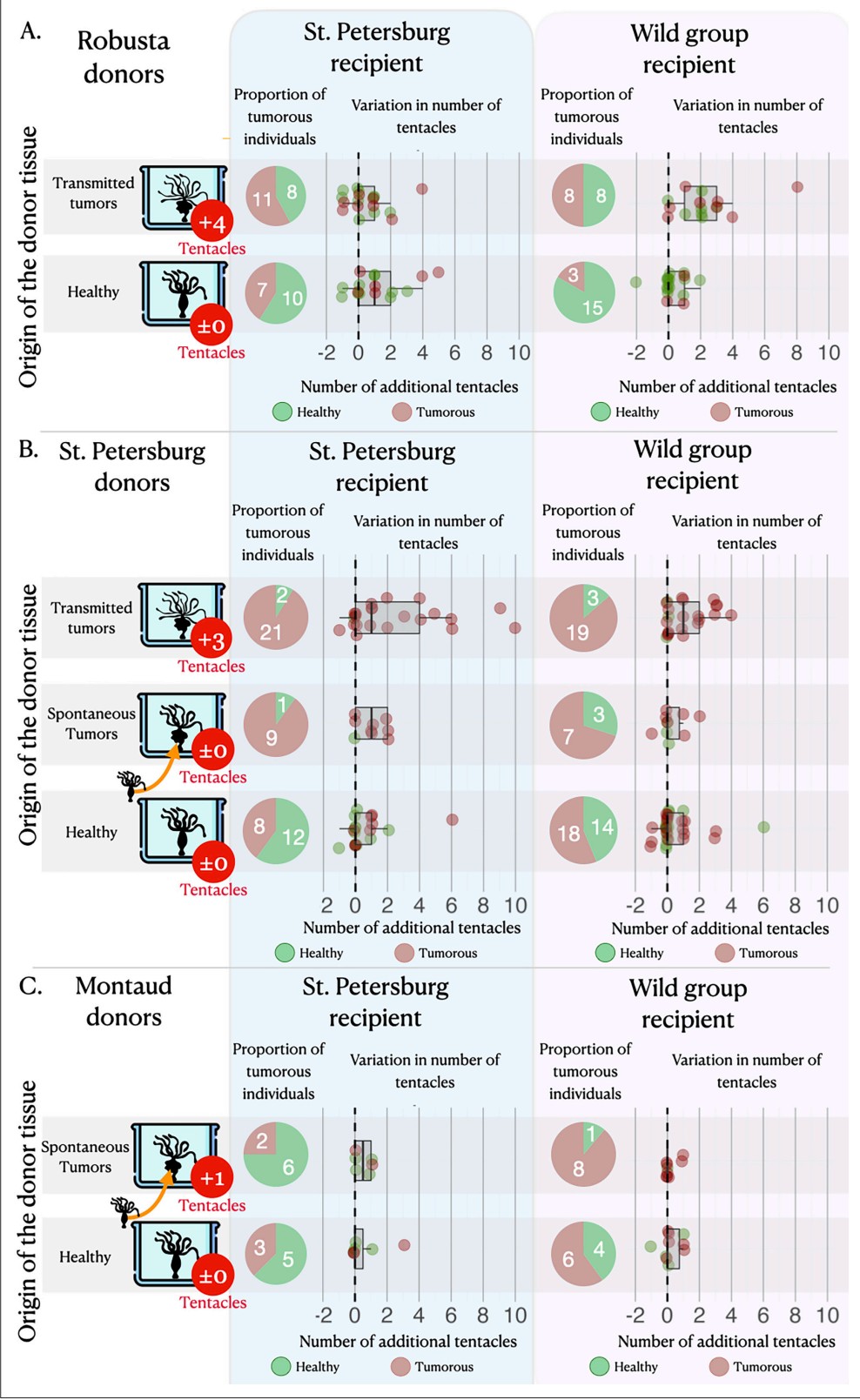

**Figure 3.** Tumor appearance and changes in tentacle number after the grafts. Three sets of matrices showing (i) the proportion of individuals (pie charts) that have developed tumors (in red) or lacked tumors (in green) and (ii) the variation in tentacle numbers (graphs) among non-tumorous individuals (green dots) and tumorous individuals (red dots), following the grafting of healthy tissue (green bars) or tumorous tissue (from spontaneous

*Figure 3 continued on next page*

*Figure 3 continued*

– orange bars, and transmitted – red bars, tumors) from Robusta, St. Petersburg, or Montaud donor lineages (left panel) onto healthy recipients from the St. Petersburg or wild groups (top panels). For the donor lineages, the arrows indicate that the individuals with spontaneous tumors were isolated from the same cultures as the healthy individuals, whereas the individuals with transmitted tumors represent a distinct lineage; the median number of supernumerary tentacles in the tumorous donor individuals are also shown (in red circles).

tissues (*Figure 3*, pie charts; 48.75, 48.12, and 28.92%, on average, for healthy tissues from Mt, Spb, and Rob, respectively).

## Development of new tentacles following tissue transplantation

The number of tentacles (*Figure 3A*, dot plots) showed a significant increase in individuals transplanted with tissue from transmitted tumors (even if the effect remains low when the donor is Rob and the recipient is SpB), whereas there was no significant effect in individuals transplanted with tissues from spontaneous tumors (*Figure 3B and C*, dot plots), relative to individuals transplanted with healthy tissue. On average, an individual transplanted with transmitted tumor tissues developed 1.45 [1.21–1.73] additional tentacles while individuals transplanted with healthy tissue developed only one additional tentacle (IRR = 1.45 [1.21–1.73], p<0.001), with no significant effect of the donor lineage. Furthermore, we also detected a negative effect of the initial number of tentacles of the recipient (*Figure 3*, dot plots): when the recipient already had a relatively high number of tentacles, the increase in the number of tentacles after transplantation was reduced (IRR = 0.82 [0.74–0.92], p=0.001), highlighting a possible threshold effect. In contrast, we do not detect any significant effect of the number of tentacles of the recipient on the number of tentacles developed after the graft of tissues from spontaneous tumors (see model selection in supplementary section 4).

## Relationship between budding rate, number of supernumerary tentacles, and tumor development

Regardless of the type of donor and recipient, the number of buds produced per individual during the 10 weeks of the study was found to be significantly correlated with the number of supernumerary tentacles developed by the individual during the same period (*Figure 4*). However, the slope of this correlation was notably reduced for hydras with tumors (*Figure 4*, n=92, estimate = –2.75 [–5.21–0.29], p=0.029). Consequently, all other factors being equal, an individual developing a tumor will require more supplementary tentacles than a healthy individual to achieve the same number of buds produced. Noticeably, we observed that the impact of the number of tentacles and tumor development on the number of buds produced became statistically detectable only after the seventh week of the experiment.

## Discussion

Many parasites from different taxa (e.g., viruses, bacteria, protozoa, helminths, fungi, arthropods) have evolved the ability to manipulate the phenotype of their host to favor their own transmission and/or survival (*Hughes et al., 2012*). A wide variety of host traits can be modified by parasites, including reproduction, behavior, color, morphology, and/or physiology (*Barnett, 1983*; *Dawkins, 1999*; *Agnew et al., 2000*). Although theory predicts that transmissible cancers should evolve a true parasitic lifestyle over time (*Dujon et al., 2020*; *Burioli et al., 2021*), to date there has been no evidence of host phenotypic manipulation by transmissible tumor cell lines (*Tissot et al., 2016*).

In their pioneering work on transmissible tumors in hydra, *Domazet-Lošo et al., 2014* mentioned the existence of an increased number of tentacles in the tumor-bearing lineages that have been propagated in the laboratory, but the precise cause of this phenotypic change was not understood. Here, we first highlight that this is not a general attribute of tumor-developing hydras since both Mt and SpB polyps developing spontaneous neoplasms do not show a similar large increase in the number of tentacles (*Figure 2A*). These results support the hypothesis that spontaneous tumors, which have no evolutionary history with their host, do not possess (or just to a small extent) the ability to influence its phenotype. Conversely, our results showed that transmissible tumors propagated in the laboratory for more than 10 years can induce the growth of additional tentacles even when they

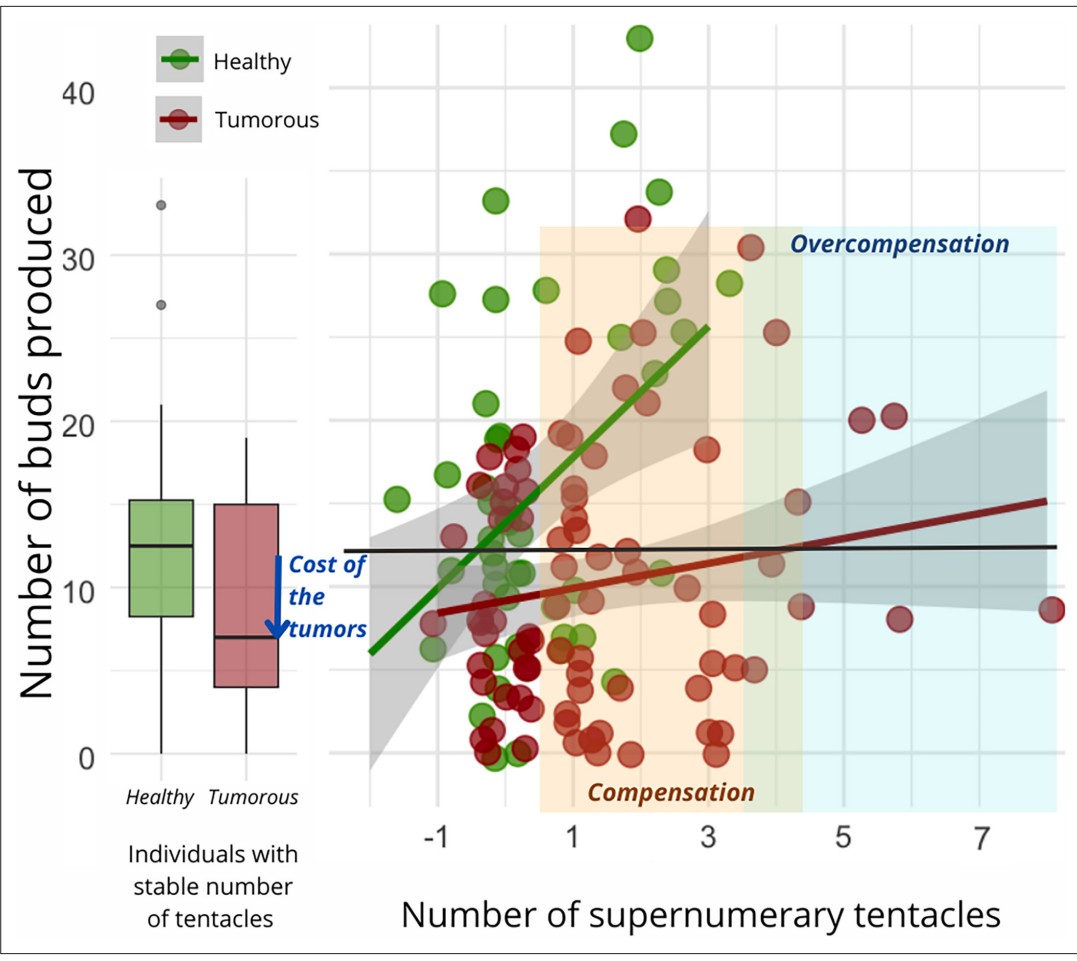

**Figure 4.** Number of buds produced depending on the development of supernumerary tentacles. On the left, the box plot exhibits the bud production of healthy (n=47, green) and tumorous individuals (n=28, red) with a constant number of tentacles during the 10-week monitoring post-grafting. The arrow signifies the reduction in bud production attributed to tumor costs. On the right, each point represents bud production for an individual (healthy in green, n=83, and tumorous in red, n=116) based on the number of tentacles developed in the 10 weeks following the transplantation. The orange zone represents tentacle-induced compensation for tumor costs compared to healthy individuals with a stable number of tentacles, while the blue zone indicates overcompensation beyond expected values for a healthy individual. The interaction between tentacle number and tumor occurrence significantly explains bud production variation (random linear mode, estimate = –2.75, CI: –5.21 to –0.29, p=0.029).

are transplanted in new hosts (even if this effect seems low with SpB grafted with tumors from Rob) as they seem to do in their original host lineage. In addition, while transplantation of spontaneous tumors never promoted an increase in tentacle numbers, transmissible tumors did, regardless of the origin of the recipient hydra (although one may suspect a different pattern between recipient strains that may not be detected due to a lack of power). These results are not consistent with the hypothesis that the development of additional tentacles could indicate a compensatory response of tumor-bearing polyps, for example, to better cope with resources redirected to growing tumors. Similarly, in agreement with previous work showing that the tumor phenotype can be observed in tumor-bearing hydras that do not consistently have an altered microbiome (this is the case for SpB, but not Rob; *Domazet-Lošo et al., 2014*), our transplantation experiments do not support the hypothesis that the alteration in tentacles' number could be caused entirely by bacteria. Indeed, both the SpB and Rob tumors were able to trigger supernumerary tentacles in their new hosts, even if Rob microbiome is the same between tumorous and non-tumorous hydras (*Domazet-Lošo et al., 2014*). Thus, although the exact mechanisms promoting the growth of additional tentacles remain to be identified, this is

the first evidence of transmissible tumor cells that have evolved the ability to manipulate an external phenotypic trait in their host. Further experiments focusing on transcriptional and proteomic changes (*Takahashi et al., 2005*) will certainly improve our knowledge of these phenotypic manipulations.

While it seems clear that the growth of supernumerary tentacles is induced only by the presence of transmissible tumor cells, to argue for host manipulation in the strict sense (as observed in the case of manipulative parasites; see *Ewald, 1980*; *Poulin, 2010*) it is necessary to demonstrate that this phenotypic change is indeed associated with an increase in tumor transmission. One possibility is that the supernumerary growth of tentacles, by increasing the rate of prey capture, merely compensates for the pathological costs associated with the presence of the tumor, the growth of which requires additional resources. Under this hypothesis, tumorous hydra with supernumerary tentacles would not necessarily exhibit an increased budding rate as it would be more indicative of a host manipulation aimed at better tolerating the tumor (i.e., compensating for vital resources diverted toward the tumor; *Thomas et al., 2020*). It is also possible than the growth of supernumerary tentacles, through a higher feeding rate, enhances the host's budding rate and, consequently, the vertical transmission of the tumor. Our results support the hypothesis that tumors do indeed impose a cost on the reproductive potential of the host in terms of the weaker budding of tumoral individuals, but that this cost can be offset by the growth of supernumerary tentacles. Beyond this first conclusion, our data also support the hypothesis that the supernumerary growth of tentacles can sometimes not only compensate for the cost of the tumor, but also leads for some cancerous individuals to a significant increase in the number of buds, beyond the values presented by healthy individuals with a normal number of tentacles. Given that the transmission rate of tumors remains constant regardless of the budding rate (*Boutry et al., 2022b*), all our results thus suggest that tumor-induced supernumerary tentacle growth has likely evolved as an adaptation that enhances their transmission. Further research should focus on underlying the bio-physiological pathways and energetical costs of this manipulated phenotype (*Takahashi et al., 2005*; *Sebens et al., 2017*).

An intriguing finding from the present study is that in *H. oligactis* new tumors can develop following injury and/or transplantation of healthy tissue alone. Additional experiments are required to formally describe these outgrowths as tumors (although there are strong phenotypic similarities with spontaneous tumors; see *Domazet-Lošo et al., 2014*) and to understand the cause and mechanisms underlying their development. We can speculate that the act of grafting is likely to weaken the hydra's ability to control abnormal cell proliferation and/or make them more susceptible to dysbiosis known to promote tumorigenesis (*Rathje et al., 2020*). In any case, despite the appearance of spontaneous tumors even after transplantation of healthy tissue, it is striking that only transmissible tumors are able to consistently alter and increase the number of tentacles in the transplanted individuals, which ultimately reinforces our conclusions.

An alternative explanation for our results would be that the increase in tentacle number following transplantation is a response of the hydra when non-self-tissues are introduced into the polyps. This would imply that the tumors, owing to their own evolution over years of vertical transmission, have diverged so significantly that they are perceived as non-self by the recipient hydras. However, we do not favor this hypothesis since when *Hydra vulgaris* are grafted with tissue from *H. oligactis*, we simply observe a destruction of the non-self-tissue (*Bosch and David, 1986*). In contrast, there is evidence of a lack of intra-specific non-self-recognition in *H. vulgaris* (*Kuznetsov and Bosch, 2003*), without any reporting of growth of additional tentacles. Also, since healthy and spontaneously tumorous hydras have been shown to sometimes grow a few additional tentacles, it would be necessary to explore in the future the extent to which the responsible environmental factors are similar to those potentially produced/exploited by transmissible tumors (see, for instance, *Lefèvre et al., 2008*). Further work, including experimental evolution, would be needed to assess the extent to which tentacle growth promotion is a capability that could be acquired over time from spontaneous tumor cells at least for those that also acquired the capacity to be transmitted.

## Materials and methods
### Biological material
All hydras were maintained according to standard rearing protocols (*Boutry et al., 2022a*). Three different main lineages of *H. oligactis* were used in our study. The St. Petersburg (named thereafter

SpB) and the Robusta (named thereafter Rob) strains were obtained from Thomas Bosch's laboratory in Kiel (*Domazet-Lošo et al., 2014*). For each of these two lines, control lines were also isolated from healthy polyps collected during the same sampling; these control groups do not harbor any sign of transmitted tumors and are supposed to be genetically identical to their respective tumorous lines (*Boutry et al., 2022b*). A third tumorous lineage has been recently established in Frédéric Thomas's laboratory in France and was isolated from a polyp sampled in Montaud water pond (43.748211°N, 3.989871°E); this lineage, named Montaud (thereafter Mt), was maintained in the lab since August 2021 (*Boutry et al., 2023*). To date, we have been unable to establish a distinct control and vertically transmitted tumorous lineage for this new strain due to variable transmission rates of these tumors and the sporadic occurrence of spontaneous tumors from initially healthy-looking polyps. In addition, a group named 'wild group' was used as recipient of the transplanted tissues; this group was composed of randomly sampled *H. oligactis* and assumed to be genetically diverse and distinct from the SpB and Rob lineage. These wild hydras were also sampled in the Montaud pond, between 15 February 2022 and 15 April 2022.

In this study, two types of tumors were distinguished. 'Transmitted tumors' correspond to those from the tumorous hydra lineages (SpB and Rob) obtained from Thomas Bosch's laboratory in Kiel; hydras bearing these tumors possess an enhanced number of tentacles and these tumors have been vertically transmitted through budding for 15 years (*Lange, 2010*). 'Spontaneous tumors' correspond to tumors that sporadically appear in Mt lineage, but also among healthy polyps of the SpB control lineage (which were isolated from the common culture as soon as they appeared). Noticeably, tumors were never observed to appear spontaneously in the Rob control lineage.

## Grafting protocol

The transplantation experiment was performed according to *Lenhoff, 2013*; *Dujon et al., 2022* using healthy and tumorous donors of the SpB, Rob, and Mt lineages. For the recipients, we used healthy individuals of the SpB lineage and the wild group (see *Figure 1* for an illustration of the experimental design). Since transplants on healthy Rob recipients led to massive mortality, we excluded them from this analysis. The recipients from the 'wild group' (because they may be genetically diverse) were cut in half longitudinally 1 week before transplantation and allowed to fully regenerate in order to obtain two genetically identical individuals on which to perform the control and tumor transplants. The manipulations were performed under a binocular magnifying glass (Olympus model SZ51, magnification from 0.67× to 4.5×) and the tools were disinfected with 90% alcohol and rinsed with Volvic before and after each transplant. Donor and recipient hydras were starved for 24 hr prior to transplantation to prevent digested food remaining in the gastric cavity from being released during grafting.

Transplantations were realized in a glass Petri dish containing 150 mL of Volvic. Using a flat scalpel, the head of the donor hydra was cut and removed from the beaker. Two cuts were made in the body of the hydra, resulting in a ring-shaped graft (*Figure 1*). Using fine forceps, a piece of fishing line (nylon, diameter 0.28 mm) was inserted into the center of the graft. A small incision was made halfway up the recipient hydra, and the fishing line that has been previously inserted into the graft was then passed through this incision and exited through the mouth of the recipient hydra to limit tissue damage. The graft was held to the recipient by sections of polyethylene tubing (*Figure 1*; internal diameter 0.30 mm, Sensa). The fishing line and the tubes were removed 3 hr after the graft, and the experimenter could then check the acceptance of the graft. The hydras whose graft was not accepted were eliminated and the grafting was repeated (see *Dujon et al., 2022* for a video of the grafting steps). Three days later, the protrusion created by the graft was excised using a scalpel blade, allowing the individual to return to a normal morphology. The period of 3 days was assumed to be sufficient time to allow tumor cells to migrate into the tissue before graft removal (*Domazet-Lošo et al., 2014*). Freshly transplanted hydras were deprived of food for a week to avoid water pollution from feeding during regeneration, then returned to a normal diet on day 8 post-transplant.

## Temporal follow-up and statistical analysis

The number of tentacles in donor and recipient individuals was assessed before the transplantations. The individuals with transplants were then followed weekly for 70 days to record the number of tentacles, the date of tumor onset, the asexual reproduction through bud release, and the date of eventual death. Detached buds from the grafted individuals were removed after each feeding. Generalized

linear mixed-effects models were used to quantify the impact of donor and recipient characteristics on the different traits observed in grafted individuals. For each analysis, the most parsimonious model was selected following the method used by *Zuur et al., 2009*. The use of Akaike's conditional criterion (AICc), therefore, allows us to exclude some irrelevant effects and present here only the results of the most relevant models (see supplementary). All the models, their effects (fixed and random), and their respective AICc are presented in the online supplementary material. The absence of overdispersion and the balance of the residuals were also checked using the DHARMa package (*Xie et al., 2021*). Analyses were performed using R software (version 4.1.3) (*R Development Core Team, 2022*) and various packages (see the exhaustive list in Appendix). Because of the unbalanced distribution of tumor-type modalities within the different lines (transmissible tumors in SpB and Rob, spontaneous tumors in Mt and SpB), we had to separate all analyses into two parts, one including all lines with transmissible tumors as well as their controls, and one including all lines with spontaneous tumors and their controls. Random effects of transplantation date and/or batch were evaluated and included in the models for which they were relevant.

The median number of donor tentacles was compared using Wilcoxon tests between the tumorous and the control groups. Mortality rates during the experiment were compared using a random generalized model with a binomial distribution (see the analysis and results in supplementary section 2). The rate of tumor development was modeled using a random generalized model with a binomial distribution (see the analysis and results in supplementary section 3). The different models constructed evaluated the fixed effects of donor lineage (SpB, Rob, or Mt), donor status (tumor or healthy), and recipient lineage (SpB or Wild group).

The change in the number of tentacles was then analyzed using a delta variable: the difference between the maximum number of tentacles developed during the study and the number of tentacles of the recipient on the day of transplantation (see the analysis and results in supplementary section 4). This variable was then modeled according to a random generalized models with a Poisson distribution (we turned all values into positive values by simply adding 2 to all values). In the models, several fixed effects were tested: tumor status and lineage of the donor, lineage of the recipient, and initial number of tentacles of the recipient and donor. The impact of tumor development after transplantation on the number of tentacles was excluded because of a confounding effect between donor status and tumor development (see in supplementary).

The relationship between the number of buds produced and the number of tentacles developed was subsequently examined on the complete dataset (transmissible and spontaneous tumors together) utilizing the same delta variable (refer to the analysis and results in supplementary section 5). The number of buds produced during the 10-week observation period was then subjected to modeling through random linear models. In these models, various fixed effects were assessed, including the tumor status of the grafted individuals, the delta variable representing the number of additional tentacles, and the group consisting of the lineage and cancerous status of both the donor and recipient. The latter was treated as random effects to emphasize our focus on intra-group variations in drawing conclusions. Figures and tables were created with the packages GGplot2 (*Wickham, 2016*), Sjplot (*Lüdecke, 2023*), ggstatsplot (*Patil, 2021*), and the software Keynotes.

## Acknowledgements

We are grateful to Thomas Bosch's laboratory, and in particular to Alexander Klimovich for providing the first hydras, methods, and ideas on which the authors first met and built their research. This work was supported by the MAVA Foundation, the Hoffmann Family, and the ANR EVOSEXCAN to FT. JB's salary was funded by the University of Montpellier.

## Additional information

### Funding

| Funder | Grant reference number | Author |
| --- | --- | --- |
| University of Montpellier | | Justine Boutry |

| Funder | Grant reference number | Author |
|---|---|---|
| MAVA Foundation | | Fréderic Thomas |
| Agence Nationale de la Recherche | EVOSEXCAN | Fréderic Thomas |

The funders had no role in study design, data collection and interpretation, or the decision to submit the work for publication.

## Author contributions

Justine Boutry, Conceptualization, Data curation, Formal analysis, Supervision, Investigation, Visualization, Methodology, Writing – original draft, Writing – review and editing; Océane Rieu, Investigation, Visualization, Writing – review and editing; Lena Guimard, Investigation, Methodology, Writing – review and editing; Jordan Meliani, Resources, Investigation; Aurora M Nedelcu, Beata Ujvari, Conceptualization, Supervision, Funding acquisition, Writing – review and editing; Sophie Tissot, Conceptualization, Formal analysis, Investigation, Methodology, Writing – review and editing; Nikita Stepanskyy, Writing – review and editing; Rodrigo Hamede, Conceptualization, Funding acquisition, Writing – review and editing; Antoine M Dujon, Formal analysis, Supervision, Validation, Writing – review and editing; Jácint Tökölyi, Conceptualization, Supervision, Validation, Methodology, Writing – review and editing; Fréderic Thomas, Conceptualization, Supervision, Funding acquisition, Validation, Writing – original draft, Project administration, Writing – review and editing

## Author ORCIDs

Justine Boutry ⑩ https://orcid.org/0000-0002-0707-1202
Océane Rieu ⑩ https://orcid.org/0009-0001-2341-9993
Aurora M Nedelcu ⑩ https://orcid.org/0000-0002-7517-2419
Beata Ujvari ⑩ https://orcid.org/0000-0003-2391-2988
Antoine M Dujon ⑩ https://orcid.org/0000-0002-1579-9156
Jácint Tökölyi ⑩ https://orcid.org/0000-0002-6908-6493
Fréderic Thomas ⑩ https://orcid.org/0000-0003-2238-1978

Reviewer #1 (Public review): https://doi.org/10.7554/eLife.97271.3.sa1
Reviewer #2 (Public review): https://doi.org/10.7554/eLife.97271.3.sa2
Author response https://doi.org/10.7554/eLife.97271.3.sa3

# Additional files

## Supplementary files
MDAR checklist

## Data availability
This repository contains complete dataset, details of the performed analysis and supplementary analysis at https://justineboutry.github.io/Supplementary.io/.

The following dataset was generated:

| Author(s) | Year | Dataset title | Dataset URL | Database and Identifier |
|---|---|---|---|---|
| Bountry J | 2023 | Hydra graft dataset | https://justineboutry.github.io/Supplementary.io/ | GitHib, justineboutry.github.io/Supplementary.io |

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
