## [Editor Report · eLife Assessment]

This interesting study explores whether tumor cells can manipulate their *Hydra* hosts and includes **important** findings on the consequences for the fitness of the host *Hydra*. The evidence supporting these findings is **convincing**. The work will be of broad interest to many fields including development biology, evolutionary biology and tumor biology.

---

## [Referee Report · Reviewer #1 (Public review)]

Summary:

In this manuscript, BOUTRY et al examined a cnidarian Hydra model system where spontaneous tumors manifest in laboratory settings, and lineages featuring vertically transmitted neoplastic cells (via host budding) have been sustained for over 15 years. They observed that hydras harboring long-term transmissible tumors exhibit an unexpected augmentation in tentacle count. In addition, the presence of extra tentacles, enhancing the host's foraging efficiency, correlated with an elevated budding rate, thereby promoting tumor transmission vertically. This study provided the evidence that tumors, akin to parasitic entities, can also exert control over their hosts.

Strengths:

The manuscript is well-written, and the phenotype is intriguing.

---

## [Referee Report · Reviewer #2 (Public review)]

Background and Summary:

This study addresses the intriguing question of whether and how tumours can develop in the freshwater polyp hydra and how they influence the fitness of the animals. Hydra is notable for its significant morphogenetic plasticity and nearly unlimited capacity for regeneration. While its growth through asexual reproduction (budding) and the associated processes of pattern formation have been extensively studied at the cellular level, the occurrence of tumours was only recently described in two strains of Hydra oligactis (Domazet-Lošo et al, 2014). Here, tumour-like tissue bulges formed within the ectodermal epithelial layer and contained increased numbers of interstitial cell-like cells which exhibited female germline markers, but none specific for somatic derivatives of interstitial stem cells (e.g., nematocytes, neurons or glandular cells). It seems likely that the cellular basis of these malformations is a misregulation of oogenesis. In wild-type polyps, interstitial-cell-related germline precursors give rise to oocytes and nurse cells, which are subsequently phagocytosed by the growing egg cell. By comparison, in the mutant strains, this uptake is disturbed, but the homeostasis between germline cells and epithelial cells must remain functional enabling further growth pattern formation in hydra. Determining whether this differentiation arrest constitutes a neoplasm also remains a challenge.

Clonal lines of both strains have been maintained in the laboratory for years and have also been used by Boutry and colleagues. They published two further papers on the ecological and evolutionary aspects of hydra tumour formation (Boutry et al 2022, 2023), which is also the focus of this manuscript. In their paper, the authors demonstrate an increase in the number of tentacles when "tumour tissue" was transplanted to intact gastric tissue of wildtype and mutant strains. While the impact on tentacle formation is relatively modest, small, it indicates a potential influence on the cross-talk between epithelial and interstitial cells in growth control (proportion regulation). The presented data are of interest, although the underlying molecular processes remain to be demonstrated. The authors offer a different interpretation. They conclude that this growth pattern (increased number of tentacles) is correlated with "reducing the burden on the host by (over-) compensating for the reproductive costs of tumours" and claim that "transmissible tumours in hydra have evolved strategies to manipulate the phenotype of their host".

Strength

The question of whether and how tumours can develop in simple systems, here the freshwater polyp hydra, is of general interest. The authors describe transplantation experiments by using mutant strains that indicate an influence of tumour-like malformation on pattern formation. The experiments also suggest an interaction between epithelial cells and germline cells during oogenesis, interfering with the homeostatic growth control between the cell lineages.

Weaknesses

Although it is stimulating to consider a fresh perspective from other disciplines (here, ecological and evolutionary aspects), it appears that this interpretation of the data (reducing the burden on the host by (over-) compensating for the reproductive costs of tumours) is somewhat beyond what can be reasonably inferred from the evidence presented. It is essential, particularly in the context of evolutionary biology, to conduct further analysis of the underlying cell biology of these intriguing mutant hydra strains. Such cellular analysis is a relatively straightforward approach that could provide a mechanistic understanding of the phenomenon described by the authors.

---

## [Author Response]

The following is the authors’ response to the original reviews.

**Public Reviews:**

**Reviewer #1 (Public Review):**
Summary:In this manuscript, BOUTRY et al examined a cnidarian Hydra model system where spontaneous tumors manifest in laboratory settings, and lineages featuring vertically transmitted neoplastic cells (via host budding) have been sustained for over 15 years. They observed that hydras harboring long-term transmissible tumors exhibit an unexpected augmentation in tentacle count. In addition, the presence of extra tentacles, enhancing the host's foraging efficiency, correlated with an elevated budding rate, thereby promoting tumor transmission vertically. This study provided evidence that tumors, akin to parasitic entities, can also exert control over their hosts.Strengths:The manuscript is well-written, and the phenotype is intriguing.Weaknesses:The quality of this manuscript could be improved if more evidence were to be provided regarding the beneficial versus detrimental effects of the tumors.

We thank the reviewer for taking the time to examine our work carefully and for their highly relevant comments and precise suggestions. We have incorporated these suggestions, which greatly improved the clarity of our manuscript concerning the beneficial and detrimental effects of tumors. Specifically, we have added a new analysis and rephrased the results section, as well as the corresponding sentences in the discussion, to enhance clarity.

Additionally, regarding the impact of tumor size on the development of supernumerary tentacles, we have included as suggested a new analysis that was previously only available in the supplementary materials of the earlier version. This addresses the reviewer's question and significantly enhances the quality of our paper.

We have thanked the two referees in the Acknowledgements section of our article.

**Reviewer #2 (Public Review):**
Background and Summary:This study addresses the intriguing question of whether and how tumors can develop in the freshwater polyp hydra and how they influence the fitness of the animals. Hydra is notable for its significant morphogenetic plasticity and nearly unlimited capacity for regeneration. While its growth through asexual reproduction (budding) and the associated processes of pattern formation have been extensively studied at the cellular level, the occurrence of tumors was only recently described in two strains of Hydra oligactis (Domazet-Lošo et al, 2014). In that research, an arrest in the differentiation of female germ cells led to an accumulation of germline cells that failed to develop into eggs. In hydra, fertile egg cells typically incorporate nurse cells, which originate from large interstitial stem cells (ISCs) restricted to the germline, through apoptosis. However, this increase in apoptosis activity is absent in "germline tumors," and germline ISCs instead form slowly growing patches that do not compromise tissue integrity. Despite the upregulation of certain genes associated with mammalian neoplasms (such as tpt1 and p23) in this tissue, determining whether this differentiation arrest and the resulting egg patches truly constitute neoplasms remains a challenge.The authors have recently published two papers on the ecological and evolutionary aspects of hydra tumor formation (Boutry et al 2022, 2023), which is also the focus of this manuscript. They transplanted tissues derived from animals with germline tumors to wildtype animals and analyzed their growth patterns, specifically the number of tentacles in the host tissue. They observed that such tissues induced the growth of additional tentacles compared to tissues without germline tumors. The authors conclude that this growth pattern (increased number of tentacles) is correlated with "reducing the burden on the host by (over-)compensating for the reproductive costs of tumors" and claim that "transmissible tumors in hydra have evolved strategies to manipulate the phenotype of their host". While it might be stimulating to add a fresh view from other disciplines (here, ecological and evolutionary aspects), the authors completely ignore the current knowledge of the underlying cell biology of the processes they analyze.Strengths:The study focuses on intriguing questions. Whether and how tumors can develop in the freshwater polyp hydra, and how they influence the fitness of the animals?Weaknesses:Concept of germline tumors.The conceptual foundation of their experiments on germline tumors was the study of Domazet-Lošo et al (2014) introducing the concept of germline tumors in hydra (see above). While this is an intriguing hypothesis, there has been little advancement in comprehending the molecular mechanisms underlying tumor formation in hydra beyond this initial investigation. Germline tumors in hydra do not fully meet the typical criteria for neoplasms observed in mammalian tissues. More importantly, a similar phenotype was already reported by the work of Paul Brien and described as "crise gametique" (Brien, 1966, Biologie de la reproduction animale - Blastogenèse, Gamétogenèse, Sexualisation, ed. Masson & Cie, Paris). This phenomenon of gametic crisis is unique to Hydra oligactis, a stenotherm, cold-adapted cosmopolitan species. In this species, gametogenesis severely impacts the vitality of the polyps, often leading to complete exhaustion and death (Tardent, 1974). Animals can only be rescued during the initial phase of the cold-induced sexual period see also the research of Littlefield (1984, 1985, 1986, 1991). The observed arrest in differentiation arrest in germline tumors might represent an epigenetically established consequence of surviving gametogenesis. Regrettably, this important work was not mentioned by the authors or by Domazet-Lošo et al. (2014), highlighting a notable gap in the recognition of basic research in this area that might challenge the hydra tumor hypothesis."Super-nummary" tentacles in graft experiments.The authors describe that after grafting tissue from animals with germline tumors to wild-type animals, the number of tentacles in the host tissue increased when the donor tissue had germline tumors. A maximum effect of four additional tentacles was found with donor strain H. oligactis robusta and three additional tentacles with donor strain H.oligactis St Petersburg. In general, H.oligactis wild-type host strains had fewer tentacles than H.oligactis St Petersburg strains. This is consistent with the results of Domazet-Lošo et al (2014) who showed that the number of tentacles increased in the strains with germline tumors. What conclusions can be drawn from these experiments?The authors might want to conclude that transmissible tumors in Hydra have developed strategies to manipulate the phenotype of their host. But there is no evidence for this, as essential controls are missing. It is known that the size of hydra polyps is proportion-regulated, i.e. the number of tentacles varies with the size and number of (epithelial) cells. Such controls are missing in the experiments. There is also a lack of controls from wild-type animals in gametogenesis: it is very likely that grafts with wild-type animals with egg spots of comparable size as the germline tumors (see above) will result in similar numbers of tentacles in host tissue.

We thank the reviewer for their thoughtful comments. While we appreciate the concerns raised, we maintain that the evidence provided by Domazet-Lošo et al. (2014, Nature Communications) supports the relevance of this model, including the suggested comparisons with the expression profiles of individuals undergoing induced sexual reproduction. Our study focuses primarily on the impact of these tumors on the host phenotype rather than their origin. Tumors are defined as accumulations of abnormally proliferating cells. This includes the definition provided by the referee, which describes “apoptosis activity as absent in 'germline tumors,' with germline ISCs forming slowly growing patches.” Compromise of tissue integrity is not a criterion for defining neoplasms, and many benign neoplasms do not meet this criterion. We are interested in continuing this discussion with the referee to better understand the expected evidence and agree that histological nomenclature could be improved. While further investigation into the cell biology of these tumors would be valuable, this is currently beyond the scope of our article but is being pursued in separate research.

We also appreciate the points raised regarding the definition of germline tumors and the reference to the pioneering work of Paul Brien. However, in that publication, the concept of gametic crisis in *H. oligactis* describes reproductive exhaustion leading to death, rather than abnormal cell proliferation indicative of a tumor-like phenotype. This distinction likely explains why this specific paper was not cited previously.

Our study builds on prior research using the same model (e.g., Domazet-Lošo et al. 2014; Boutry et al. 2023) and describes observations across different hydra strains from various locations worldwide (not just two), all conducted under stable warm temperatures that are not conducive to sexual development. These investigations reveal a phenomenon distinct from the senescence observed post-reproduction in *H. oligactis*. The phenotype we describe, characterized by an accumulation of cells in the ectoderm, aligns with studies referenced by the reviewer from leading groups in hydra research, known for their expertise in hydra cellular biology. We have relied on these studies after carefully reviewing their results and receiving training from these experts. Furthermore, our team is focused on eco-evolutionary topics and does not aim to specialize in cellular biology, as other teams are already dedicated to that field.

We also thank the reviewer for their comments on the relevance of our findings and the missing controls. However, we have noted that the reviewer may have misunderstood our experimental design and results.

Firstly, it appears that the reviewer based their critique mainly on the initial sentences of our Results section (illustrated in Figure 2), which outline the donor groups used in our study rather than presenting the results of the grafting experiments. This description alone is insufficient for drawing conclusions, which is why we conducted further analyses using these donor groups grafted onto different recipients. The maximum effects mentioned by the reviewer (+10 tentacles with St. Petersburg tumoral tissue and +8 tentacles with Robusta tumoral tissue, Results Section 2) represent only a part of our study. We encourage the reviewer to focus on the model analyses presented in Results Section 2, which directly relate to the grafting experiments and provide a more comprehensive evaluation of our results and conclusions. These analyses include comparisons between transmissible tumors and spontaneous tumors, offering deeper insights into their effects on tentacle development.

In our methods (as depicted in Figure 3), we explicitly compared different types of tumorous tissue from various donors, distinguishing between spontaneous and transmissible tumors. Although we avoid labeling spontaneous tumors as "controls" to prevent confusion with healthy tissue controls, they serve as controls to the “treatment” that involves transmissible tumors, and thus are appropriate comparisons for assessing the size effect suggested by the reviewer. Spontaneous and transmissible tumors share similar size and cellular characteristics but differ significantly in the number of tentacles their hosts possess. Furthermore, we refer the reviewer to a relevant study (Ngo et al. 2021) that found no increase in tentacle numbers with larger polyps of healthy tissue. This reference has been included in the revised discussion (line 309 to 312), which now also addresses the potential effect of body size with additional explanations.

Regarding the suggestion to include controls from animals undergoing gametogenesis, we did not find evidence in the literature indicating an increase in tentacle numbers during this process in hydra. If such studies exist, we kindly request the complete references so we can include them in our discussion. Additionally, as noted in Brien's work, *Hydra oligactis* undergoing gametogenesis are known to either die or experience significant degeneration afterward. Transplanting tissue from dead or dying (and reproducing) hydras poses technical challenges and raises questions about whether any observed effects result from incomplete gametogenesis, the onset of senescence, or both. While these questions are intriguing, they fall outside the scope of our article.

In conclusion, we appreciate the opportunity to address these points and reaffirm that our study offers valuable insights into the evolutionary dynamics of interactions between transmissible tumor tissues and host phenotypes in hydra. We remain open to further discussion and welcome any additional feedback to enhance the clarity and robustness of our manuscript.

**Recommendations for the authors:**

**Reviewer #1 (Recommendations For The Authors):**
(1) If the fitness of hydra is altered in those with spontaneous tumors is the increased number of tentacles associated with those with transmitted tumors able to rescue this phenotype?

We thank the reviewer for reformulating our results. Indeed, fitness can be restored and even improved in tumorous polyps harboring supernumerary tentacles. This phenomenon, which we referred to as compensation and over-compensation in Section 3 and Figure 4, was initially discussed only in the discussion section. To improve the clarity of our manuscript, we have now specified this in the Conclusion (lines 345 to 347 and some minor rewording in the same paragraph) in the Results section (lines 284 to 286).

(2) Does the size of the tumor predict the number of tentacles formed?

We agree that this would be a valuable complementary analysis. We have conducted an analysis considering the qualitative size of the tumors (based on visual categories) and the number of tentacles, which is now included in our paper (lines 160-161; lines 193 to 198; lines 253 to 259; lines 314 - 322).

(3) Considering the mentioned association of body size with tentacle numbers for hydra, is a change in size a phenotype associated with transmitted tumors, and is such a phenotype transmittable.

All tumorous individuals, regardless of their tumor type, exhibit a swollen body. We have added a sentence in the introduction to clarify this point (line 62).

(4) Is there anything unique about the Rob population that would explain their mass mortality following transplantation? For instance, their resistance to spontaneous tumor formation? Similarly, is there a difference in transplantation success based on the type of tissue transplanted? The authors could address this point in the discussion.

It is a very old lineage described nearly 80 years ago. It is unknown whether natural populations of Robusta exist, and no reports of any male individuals have been documented. We have added a sentence in the Materials and Methods section to clarify this information (lines 98 to 102).

(5) What downsides are known about the transmittable tumors in hydra and how present are they in the grafted individuals? Are other physiological aspects such as mobility, regeneration, or sexual reproduction hindered?

Transmissible tumors have been associated with increased vulnerability to predation and alterations in life history traits, including a higher budding rate and decreased sexual reproduction. While we were unable to measure behavioral traits in this study of our grafted individuals, this is an intriguing avenue for further research. We have included this perspective in the discussion section as a concluding remark (lines 375 to 382). Thanks a lot for the suggestion of this conclusion.

(6) It is important to explore the mechanisms behind the phenotypic variation conferred by the types of tumors, whether of different lineage or transmissibility. For this purpose, RNA-Seq on the recipients seems like a good starting point.

Thanks for this suggestion, we've reworded the sentence about this perspective in our discussion to be more precise (line 320).

Boutry, Justine, Marie Buysse, Sophie Tissot, Chantal Cazevielle, Rodrigo Hamede, Antoine M. Dujon, Beata Ujvari, et al. 2023. « Spontaneously Occurring Tumors in Different Wild-Derived Strains of Hydra ». *Scientific Reports* 13 (1): 7449. https://doi.org/10.1038/s41598-023-34656-0.

Domazet-Lošo, Tomislav, Alexander Klimovich, Boris Anokhin, Friederike Anton-Erxleben, Mailin J. Hamm, Christina Lange, et Thomas C. G. Bosch. 2014. « Naturally occurring tumours in the basal metazoan {Hydra} ». *Nat Commun* 5 (1): 4222. https://doi.org/10.1038/ncomms5222.

Ngo, Kha Sach, Berta R-Almási, Zoltán Barta, et Jácint Tökölyi. 2021. « Experimental Manipulation of Body Size Alters Life History in Hydra ». *Ecology Letters* 24 (4): 728‑38. https://doi.org/10.1111/ele.13698.